# Compact Multi-Layered Symmetric Metamaterial Design Structure for Microwave Frequency Applications

**DOI:** 10.3390/ma16134566

**Published:** 2023-06-24

**Authors:** Tayaallen Ramachandran, Mohammad Rashed Iqbal Faruque, Mandeep Singh Jit Singh, K. S. Al-Mugren

**Affiliations:** 1Space Science Center (ANGKASA), Universiti Kebangsaan Malaysia, Bangi 43600, Selangor, Malaysia; tayachandran@gmail.com (T.R.); mandeep@ukm.edu.my (M.S.J.S.); 2Physics Department, Science College, Princess Nourah bint Abdulrahman University, Riyadh 11671, Saudi Arabia; ksalmogren@pnu.edu.sa

**Keywords:** effective medium ratio, left-handed metamaterial, multi-layered, microwave frequency, symmetric structure

## Abstract

Metamaterial analysis for microwave frequencies is a common practice. However, adopting a multi-layered design is unique in the concept of miniaturisation, thus requiring extensive research for optimal performance. This study focuses on a multi-layered symmetric metamaterial design for C- and X-band applications. All simulation analyses were performed analytically using Computer Simulation Technology Studio Suite 2019. The performances of the proposed metamaterial design were analysed through several parametric studies. Based on the observation, the proposed metamaterial unit cell design manifested resonant frequencies at 7.63 GHz (C-band) and 9.56 GHz (X-band). Moreover, the analysis of effective medium parameters was also included in this study. High-Frequency Simulation 15.0 and Advanced Design System 2020 software validated the transmission coefficient results. Simultaneously, the proposed multi-layered metamaterial design with Rogers RO3006 substrate material exhibited a unique transmission coefficient using double, triple, and quadruple layers. The two resonant frequencies in the unit cell design were successfully increased to three in the double-layer structure at 6.34 GHz (C-band), 8.46 and 11.13 GHz (X-band). The proposed unit cell design was arranged in an array structure to analyse the performance changes in the transmission coefficient. Overall, the proposed metamaterial design accomplished the miniaturisation concept by arranging unit cells in a multi-layer structure and possesses unique properties such as a highly effective medium ratio and left-handed characteristics.

## 1. Introduction

The term metamaterial refers to man-made artificial materials that are not found in nature. Metamaterials have the potential to substantially enhance our ability to manipulate interactions with electromagnetic (EM) radiation by broadening the design space and allowing the implementation of peculiar electromagnetic properties such as negative permittivity, negative permeability, and negative index of refraction. The EM constitutive properties of a structure can also be precisely controlled using metamaterials over a volumetric region, a feature exhibiting unique advantages for complex media design, such as the transformation of optical media for EM cloaking. Vibration attenuation, diffraction-limited imaging, wireless power transmission, gradient index, perfect absorbers, specific absorption rate reduction, coding metamaterial, and other innovative ideas have all resulted from the recognition of manipulating material properties through the structure [1,2,3,4,5,6,7,8,9].

Numerous studies on metamaterial design have successfully achieved improved outcomes in the proposed application fields. For instance, Hasan et al. [10] proposed a compact double-negative metamaterial with a negative refractive index bandwidth of more than 3.6 GHz. Therefore, in this work, a frequency range from 2 to 14 GHz was considered. The two arms of the introduced unit cell were separated to form a Modified-Z-shaped structure of the FR-4 substrate material. Meanwhile, Hasan and Faruque [11] proposed a left-handed metamaterial for multiband applications using azimuthal (xy-plane) angular rotations. The square resonators were divided, and a metal bar was placed in an arrangement that resembles an angular sensitive z-shaped split-ring resonator structure. A novel biaxial double-negative metamaterial for rectangular EM cloaking was introduced by Islam et al. [12] in 2015. In microwave applications, the metamaterial exhibited double-negative features along the three major axes, namely the x-, y-, and z-axes. In another study, Mishra et al. [13] explored a rectangular split-ring resonator-based gradient refractive index metamaterial structure for a lens antenna. Incorporating splits in the rectangular rings to improve the coupling of currents between the rings was the novelty of this proposed design. In 2015, Liu et al. [14] proposed a new type of tunable meta-atoms using liquid metal embedded in a stretchable polymer focusing on reconfigurable metamaterials. All flexible materials that were compatible with the surface of an interaction object were used to create the meta-atom.

On the other hand, Hasan et al. [15] introduced an enhanced dual-band square-Z-shaped double-negative meta-atom for C- and X-band applications. In this study, the meta-atom was split to appear as a square-Z-shaped structure printed on an epoxy resin fibre substrate. Tamim et al. [16] developed a horizontally inverse double L-shaped metamaterial for triple-band applications with negative permittivity and permeability at resonant frequencies. Moreover, a square split-ring resonator made of Rogers RT5880 dielectric substrate material was etched on the outer edge of the horizontally inverse double L-shaped structure. A complementary split-ring resonator-based metamaterial was constructed and explored for microwave applications with an effective medium ratio by Almutairi et al. [17] in 2019. Moreover, Yacine et al. [18] investigated a dual-band bandpass filter inspired by a pair of square-coupled interlinked asymmetric tapered metamaterial resonators for X-band microwave applications. The designed filter, consisting of a square-shaped coupled interlink, was fed in parallel mode by two microstrip lines and etched on the upper face of the used substrate.

Recently, coding metamaterial has also been used in microwave frequency applications. In 2014, Cui et al. [19] introduced the concept of digital metamaterials in two steps: coding metamaterials and unique metamaterial particles. Two distinct unit cells were utilised in the 1-bit coding scheme to simulate ‘0’ and ‘1’ elements. This study also explored the 2-bit coding metamaterial that utilised four different unit cell types, including the ‘00’, ‘01’, ‘10’, and ‘11’ elements. Meanwhile, Moeini et al. [20] proposed fractal coding metamaterials for reflective metasurfaces with self-similar pseudo-random phase responses. These responses were based on the coding strategy using fractal interpolation functions. In another study, Cuong et al. [21] constructed, simulated, and measured a metamaterial microwave absorber based on broadband coding. The analysis performed full-wave finite integration simulations through a full-sized configuration. In 2019, Zhang et al. [22] proposed a bifunctional digital coding metamaterial to engineer the propagation behaviours of acoustic and EM waves independently and simultaneously. Four rigid pillars with various material properties were employed as 1-bit reflection-type digital metamaterials with antiphase responses in both the frequency spectra as they allow independent field control.

In short, the primary challenge for researchers in this area is to create a compact metamaterial structure for microwave frequency applications to gain desired responses. This study aims to numerically analyse the scattering and effective medium parameters using several parametric elements, such as unit cell selection process analysis, multi-layered metamaterial analysis, and convention array structure exploration. This optimisation procedure adopted a simple but effective trial and error method to identify the enhanced behaviours likely to have multiple resonant frequencies and unique left-handed characteristics. After the selection process for unit cell design from four distinct metamaterial structures, the design went through extensive multi-layer analysis by adopting two types of substrate materials. Finally, the unit cell was arranged in a conventional array cell structure to explore the performance changes. Therefore, in this study, the S21 values of the proposed design were validated using HFSS 15.0 software. The data obtained were further verified using ADS software by building an equivalent circuit model. These analyses added innovation to this work because previous studies on metamaterial design with a frequency range from 4 to 12 GHz had major design, property, and behaviour limitations. Moreover, in addition to creating and synthesising a new material, other methods can also manifest unique and novel properties. However, the modification of existing metamaterials by simply changing a few physical structures or using new methods can achieve extraordinary performances. For instance, the proposed design adopted a thinner substrate material, which possesses unique features and benefits that are ideal for multi-layer design construction. Therefore, this method typically offers significant time and cost-saving advantages during the construction of metamaterial design. Furthermore, this compact design can be easily integrated inside a device or component while the conventional array cell design structure will take up more space. Hence, this study aims to investigate the features of the symmetrical metamaterial for novel weather monitoring technology. Weather monitoring is crucial to describe the current climate, detect climate changes, and supply data for models to predict future environmental changes.

## 2. Metamaterial Design Analysis

Several unique structures, as illustrated in Figure 1a–d, were proposed during the unit cell design selection process. Furthermore, the scattering parameter (S-parameter), for instance, the transmission coefficient (S21) of each design on z-axis wave propagation, was also analysed. The linear characteristics of radio frequency (RF) electronic circuits and components are represented by S-parameters. The S-parameter matrix can calculate linear network properties such as gain, loss, impedance, phase group delay, and voltage standing wave ratio. This analysis adopted three types of substrate materials with distinct thicknesses, namely FR-4, Rogers RO4350 B, and Rogers RO3006. The substrate materials possess a tangent loss (δ) and dielectric constant (ɛ) as follows: FR-4 lossy material with µ of 0.025 and ɛ of 4.3; Rogers RO4350 B lossy material with µ of 0.0037 and ɛ of 3.66; and Rogers RO3006 lossy material with µ of 0.002 and ɛ of 6.5. The thickness of the materials was 1.6 mm, 0.168 mm, and 0.25 mm, respectively.

Overall, the S21 readings indicated similar resonant patterns in Designs 3 and 4 with Rogers RO3006 substrate material, as depicted in Figure 1c,d. Design 3 exhibited two resonant frequencies at the X-band, while Design 4 manifested two peaks at C- and X-bands, respectively. Contrarily, the FR-4 material produced a single resonant for all designs except Design 3. In Design 3, the FR-4 material yielded two resonant frequencies at the X-band with magnitude values of −21.72 and −34.22. However, the Rogers RO4350 B substrate material manifested one peak for all designs except Design 1. This type of material also localised the resonant at the X-band with reasonable magnitude values. Meanwhile, the resonants of the proposed design with Rogers RO3006 substrate material are pinpointed by a red circle in Figure 1b–d. Table 1 enlists the resonant frequencies of the four distinct metamaterial designs with three substrate material types with magnitude values of more than −15.

All the introduced designs yielded unique S21 readings, indicating that they can be utilised for various applications. For instance, Design 2 with Rogers RO4350 B substrate material with a resonant frequency of 10.47 GHz and a magnitude value of −41.55, and Design 2 with FR-4 substrate material with a peak at 8.30 GHz and a magnitude value of −29.42 can be used in the stealth technology application field. However, this study aims to produce a few resonant bands by adopting a single-unit cell metamaterial. Meanwhile, the response is further increased by the arrangement of the unit cell in a multi-layered structure. Therefore, Design 4 was selected for further analysis as it adopted Rogers RO3006 substrate material.

### 2.1. Unit Cell Design

Figure 2a,b illustrate the proposed symmetric metamaterial design’s numerical simulation set up and the orthographic view from the top. The metamaterial design adopted a 0.25 mm thick substrate material known as Roger RO3006, with a dielectric constant of 6.5 and tangent loss of 0.002. Moreover, a 0.035 mm thick copper material with a conductivity (σ) of 5.80 × 10^7^ S/m was adopted to construct the metamaterial structure on the substrate material. The whole dimension of the proposed substrate material design was 5 mm × 5 mm. The unit cell design consisted of two parts aligned symmetrically to form one entire structure, as illustrated in Figure 2b. Each part comprises three rings with a width of 2.0 mm for the first two rings and 1.5 mm for the third ring. There was an approximate gap of 0.20 mm before the first ring was built. The first and last rings were split by a 0.20 mm gap in each part. A few connecting bars with a 0.20 mm width were added to the design, as depicted in Figure 2b. The dimensions of the suggested symmetric metamaterial design are summarised in Table 2. The square-shaped ring resonator was chosen because it enables the current to flow from one ring to another through the inter-ring space when exposed to an external magnetic field. It also has a higher current distribution and a slower resonant frequency. Moreover, it features an additional capacitive coupling to the composite structure, allowing a more robust resonant behaviour.

All numerical simulations were performed using Computer Simulation Technology (CST) Studio Suite 2019, a high-performance 3D EM analysis software. This computer program provides significant product-to-market advantages, including virtual prototyping prior to actual measurement, quicker development periods, and optimisation instead of experimentation. However, the S21 values of the proposed symmetrical metamaterial transmission coefficient were validated using the High-frequency Structure Simulator (HFSS) 15.0 and Advanced Design System (ADS) 2022 software. The CST program simulates design construction in four phases: unit cell design analysis, electric and magnetic field distribution inquiry, multi-layered structure integration analysis, and conventional array design examination. The effective medium properties of the metamaterial were explored in the unit cell design, while the S21 values were examined in the remaining design structures.

Once the unit cell metamaterial design was developed, a scattering parameter simulation was created using a tetrahedral mesh and a frequency-domain solver. Figure 2a illustrates the placement of the introduced design between the two waveguide ports. Moreover, the ports were placed on the negative and positive z-axes using Transverse Electromagnetic waves. Meanwhile, the y-axis was set to Perfect Magnetic Conductor, and the x-axis to Perfect Electric Conductor. Since this study focused on microwave frequency applications, 4 to 12 GHz range was employed. The scattering parameters of the suggested metamaterial were derived following the numerical modelling process.

The effective medium properties of the design, such as permittivity (ε), permeability (μ), refractive index (n), and impedance (*z*), were calculated using the collected data. The calculation was performed using the well-known Robust techniques in the *MATLAB R2021a software* [23,24]. Equations (1) to (4) define the retrieval equations of *z*, n, ε, and μ. The proposed design was described as an effective homogeneous medium to obtain effective permittivity and permeability values, whereas the *z* and n values were extracted from the S11 and S21 data. Contrarily, the ε, and μ values were derived using the impedance and refractive index values. The term d in Equation (2) refers to the overall thickness of the unit cell design used in this retrieval procedure, 0.285 mm. Once the scattering parameters and effective medium properties were satisfactory for the unit cell design, the S21 results were validated using HFSS software by adopting methods and techniques similar to those used in CST. Furthermore, the ADS software was also used to validate the S21 unit cell design results obtained from the CST software.
(1)$$z=\pm \surd \frac{{\left(1+S_{11}\right)}^{2}-S_{21}^{2}}{{\left(1-S_{11}\right)}^{2}-S_{21}^{2}}$$
$$n=\frac{1}{k_{0}d}\{[[In\left(e^{ink_{0}d}\right)]\prime \prime +2m\pi -i{\left[In\left(e^{ink_{0}d}\right)\right]}^{\prime }\},$$
(2)$$e^{ink_{0}d}=\frac{S_{21}}{1-S_{11}\frac{z-1}{z+1}}$$
(3)$$\varepsilon =\frac{n}{z}$$
(4)μ=nz

Figure 3a illustrates a simplified circuit model for the proposed symmetrical metamaterial design. The split gaps in this circuit design were maintained as capacitive effects, denoted by the letters C1, C2, C3, and C4. Meanwhile, the inductive effects were labelled as L1–L18, whereby the effects depended on the construction of strip lines. Due to the symmetric metamaterial being excited by the intensity of an electric field, the equivalent circuit diagram includes series capacitance in each split gap. The S21 results from the three software (Figure 3b) indicated nearly identical resonant patterns. The ADS software equivalent circuit model exhibited resonant frequencies of 7.66 and 9.56 GHz with magnitude values of −52.26 and −70.14, respectively. The differences between S11 and S21 are due to the power not being transmitted or reflected entirely. Thus, a break in the link between S11 and S21 is formed.

### 2.2. Effective Medium Parameters

The permittivity of the substrate material can impact the results as it can influence the resonant frequencies. When the permittivity value is increased, the peak resonant points shift to a lower frequency. Moreover, increasing the dielectric constant raises the capacitance value between the ground and the radiating elements. The reaction and response of the metamaterial design were determined by effective parameters in a medium, especially when the parameters encountered external time-varying electromagnetic fields. The interplay between the electric and magnetic fields is denoted as the physical behaviour of the EM field produced in space by time-varying electric charges. In essence, static charges only generate static electric fields in space. Meanwhile, magnetic fields can rise due to time-varying electric bills, which produce time-varying electric fields. Most materials are lossy and dispersive, resulting in complicated and frequency-dependent permittivity and permeability values. Controlling the sign of real permeability and permittivity components will generally manifest unique electromagnetic characteristics.

The effective medium parameters, namely permittivity, permeability, refractive index, and impedance of the proposed unit cell metamaterial, were examined as illustrated in Figure 4a–d. Based on the observation, negative permittivity occurred at both resonant bands, as illustrated in Figure 4a. For instance, frequency ranges from 7.71 to 8.76 GHz and 9.29 to 10.30 GHz possess negative behaviours with peaks at 7.76 and 9.56 GHz and amplitude values of −2.12 and −1.26, respectively. The proposed symmetrical metamaterial design also exhibited three permeability resonant peaks below zero amplitude, at 7.10 to 7.74 GHz, 7.75 to 8.76 GHz, and 10.25 to 12.00 GHz, as demonstrated in Figure 4b. Contrarily, permeability behaviours ranged from −14.88 to −9.62, −10.10 to −3.05, and −0.55 to −2.36, with peaks at 7.62, 8.69, and 10.47 GHz, respectively. As for the refractive index, the proposed design yielded a negative behaviour across the adapted frequency range. However, two acceptable curves were visible in Figure 4c. Meanwhile, Figure 4d indicated the impedance values at three peak points, at 7.45, 8.69, and 10.47 GHz, with amplitude values of 18.30, 27.55, and 8.63, respectively.

One of the unique characteristics of the proposed design in this study was the left-handed behaviour that occurred at the C- and X-bands. The proposed design exhibits two negative behaviour ranges of ε, μ, and n values, for instance, from −0.001 to −0.04, −7.63 to −19.70, −0.35 to −0.68 and −0.19 to −11.22, −10.65 to −2.36, −5.76 to −5.18 at 4.37 to 7.46 GHz and 7.71 to 12.00 GHz, respectively. Additionally, the effective medium ratio (*EMR*) was determined to estimate the compactness of the metamaterial design. The *EMR* was calculated using Equation (5) based on the wavelength-to-dimension ratio of the metamaterial structure. *EMR* values above 4 are ideal, as the proposed design enables negative permittivity and/or permeability behaviour. As such, the *EMR* values for both resonant frequencies were 7.86 and 6.28.
(5)$$EMR=\;\frac{Wavelength\;(\lambda )\;}{Unit\;Cell\;Length\;(L)}$$

### 2.3. Field Distribution

Figure 5a–d lists the proposed unit cell structure’s electric and magnetic field distributions at 7.63 and 9.56 GHz in two views, frontal and back. The physical properties of the EM field govern the reaction of the metamaterial structure’s field distributions. The phenomenon is feasible in space because of the time-varying behaviour of the electric charges. Furthermore, the lossy and dispersive properties of the materials can easily yield extraordinary EM properties and desired resonant frequencies. The electric field distribution was examined with a limit of 20,000 V/m, whereas the magnetic field distribution was investigated with a limit of 50 A/m. At 7.63 GHz, the unit cell design exhibited a significant electric field distribution for both sides. Meanwhile, the lowest electric field distribution occurred at the second resonant frequency localised on each design part’s first and third rings. Generally, a current induction produces a magnetic field when the wave passes through the rings. Therefore, the frontal surface of the design generates an irregular magnetic field pattern with a much lower distribution than the electric field.

On the other hand, the posterior of the design exhibited a nearly identical reaction, reaching magnetic fields of less than 12 A/m for both resonant frequencies. The magnetic field intensity is concentrated around both responses’ second and third rings. Meanwhile, the proposed design exposed a slightly narrowed field distribution at 9.56 GHz. This field distribution analysis revealed that the substrate material also responded to both frequencies in addition to the copper material. The main question raised by this phenomenon resulted in a one-of-a-kind answer. Even though the dielectric substrate material did not have free electrons, field distribution still occurred. However, materials made of metal have free electrons in their physical properties. Dielectric and metal-composed materials delocalised the electron oscillation, creating both distribution fields on the substrate material.

## 3. Multi-Layer Structure

The multi-layered metamaterial design was employed in this study because of the size constraint variables and to gain better performances. In other words, the conventional array cell usually takes up larger space and acts as a major constraint during the construction of the miniaturised structure. Therefore, double-, triple-, and quadruple-layered structures were chosen for this investigation while adopting two types of substrate materials: Rogers RO4350 B and Rogers RO3006. Figure 6a–f demonstrate the S21 results for all multi-layers in both substrate materials. Based on the results, both substrate materials manifested better S21 results than the unit cell designs. The four types of proposed unit cell designs explored in Section 2 were utilised in this section to investigate the performance changes. Most metamaterial designs increased the resonant frequencies, but Design 4 demonstrated apparent discrepancies compared to the proposed unit cell. Double-and triple-layered Design 4 manifested three resonant frequencies, while the quadruple-layer yielded four peaks with Rogers RO3006 substrate material. The Rogers RO4350 B substrate material increased the number of resonant frequencies from one to two for all three layers using Design 4.

Design 1 did not exhibit any resonant frequency for both unit cells using Rogers RO3006 and Rogers RO4350 B substrate materials. However, Design 1 using Rogers RO3006 material manifested two resonant peaks at all three layers. Meanwhile, the Rogers RO4350 B material with quadruple layers could only gain one resonant response at X-band. In addition to that, Designs 2 and 3 produced inconsistent responses in the adopted frequency range. For instance, Design 2, using the Rogers RO4350 B substrate material, exhibited similar resonant frequencies between the 8 to 10 GHz range for all three-layer structures. Meanwhile, Design 3 only manifested one resonant frequency under similar substrate material. Overall, the Rogers RO3006 substrate material exhibited better S21 results when the number of layers was increased where the resonant frequencies are distinguished by a red circle in Figure 6a–f. Therefore, S21 can be manipulated by integrating layer structure instead of a conventional array design. In a nutshell, a multi-layered structure increases the uniqueness of the proposed metamaterial design and can be applied in a wide range of practical applications.

## 4. Array Structure

In this analysis, four types of conventional array metamaterial design structures were adopted, 1 × 2, 2 × 2, 4 × 4, and 6 × 6. The proposed unit cell design with Rogers RO3006 substrate material was arranged accordingly to assess the performance changes in S21 results. Since this study mainly focuses on the miniaturisation concept, a maximum of 6 × 6 arrays was used. Moreover, the S21 results of these array structures were plotted in Figure 7. The results indicated that all the array structures manifested similar resonant frequencies to the unit cell design. The 1 × 2 array metamaterial design manifested the lowest difference in S21 values, increased by 0.06 GHz and reduced by 0.14 GHz for both resonant bands. However, the discrepancies increased for a larger array of cells at the first resonant frequency. The second resonant frequency at X-band revealed inconsistent changes in S21 values. The 6 × 6 array manifested the highest discrepancies, presenting approximately a 2.36 % reduction and 2.20% increment at C- and X-bands, respectively. Overall, S21 values can be manipulated by adapting the layered structure design and satisfying the size constraint using thinner substrate materials such as Roger RO3006.

## 5. Comparison

Table 3 compares previously published literature with the proposed design structure. In this table, most of the previously published works adopted FR-4 substrate material with a thickness of 1.6 mm, a dielectric constant of 4.3, and a tangent loss of 0.025. Most studies proposed unit cells with 10 mm^2^ × 10 mm^2^ dimension [10,16,25,26]. Another study [27] proposed smaller unit cell dimensions of 9.6 × 9.6, compared to other studies. In addition to [25], the rest of the investigation manifested more than one resonant frequency. On the other hand, several multi-layer metamaterials are also explored in this table [28,29,30]. The studies [28,29] explored wide-angle microwave absorption properties by adopting multilayer metamaterials. Furthermore, these works adopted a larger design structure compared to the rest of them. Meanwhile, the study [30] is composed of a 22.86 mm^2^ × 10.16 mm^2^ multi-layer metamaterial design for microwave frequency applications. Overall, the introduced designs exhibited unique behaviours: left-handed, negative index, double-negative metamaterials, and unique characteristics such as wide-angle and broadband. In a nutshell, the proposed 5 mm^2^ × 5 mm^2^ unit cell design successfully manifested double resonant frequencies, whereby the number of responses can be increased using a layered arrangement form. The conventional array cell design also tends to change the responses, but the design will be limited based on the size constraint. Moreover, the multi-layer concept typically reduces the overall dimension of the proposed design by adopting thinner substrate materials. One of the major challenges that a researcher faces when constructing a metamaterial design is gaining the desired resonant frequencies. A distinct combination of split gaps, the number of rings, and the specific gaps between the rings are the main factors that generally influence the outcomes. Therefore, instead of constructing a metamaterial design all over again to possess the desired responses, we adopted the simple but effective method of multi-layer arrangement to resolve this problem. Typically, this method can offer significant time and cost savings instead of utilising larger array cells in specific situations. Despite the miniaturised structure, the design displayed multiple resonant frequencies in the 4 to 12 GHz range. Furthermore, the proposed design also possessed unique left-handed characteristics for both resonant bands.

## 6. Conclusions

This study focuses on miniaturisation by integrating a symmetrical metamaterial structure for microwave frequency applications. Overall, this study adopted three types of analytical software: CST 2019, HFSS 15.0, and ADS software. Initially, all analytical simulations were performed using CST software. The S21 results of the proposed unit cell design were validated using the other two software. Based on the unit cell selection analysis observation, the metamaterial design structure and the type of substrate material adopted can greatly influence the S21 results. For instance, the four constructed distinct metamaterial designs exhibited unique responses in the C- and X-bands. However, the proposed symmetrical metamaterial design manifested double resonant frequencies between the 4–8 GHz and 8–12 GHz band ranges. The effective medium parameters of this design were also investigated in this study. The double-, triple-, and quadruple-layered metamaterial structures were utilised in this study by adopting two types of substrate material, namely Rogers RO4350 B and Rogers RO3006. The results indicated that the Rogers RO3006 material manifested better S21 results where the two initial resonants for the unit cell design were successfully increased to three and four responses at C- and X-bands, respectively. However, the Rogers RO4350 B substrate material with multi-layered structures exhibited two resonant frequencies, while the unit cell managed to gain only one. Comparatively, the multi-layered and conventional array metamaterial designs revealed avoidable outcomes. For instance, the array metamaterial design was only able to exhibit similar resonant patterns for all 1 × 2, 2 × 2, 4 × 4, and 6 × 6 structures compared to the unit cell design. Therefore, this phenomenon proved that the S21 values of the proposed unit cell design could be manipulated by adopting a layered structure because it is possible to gain multiple responses in lower bands with a compact design. In conclusion, this study highlighted the novelty of the high EMR value, left-handed characteristic, and compact design with multiple resonant frequencies.

## Figures and Tables

**Figure 1 materials-16-04566-f001:**
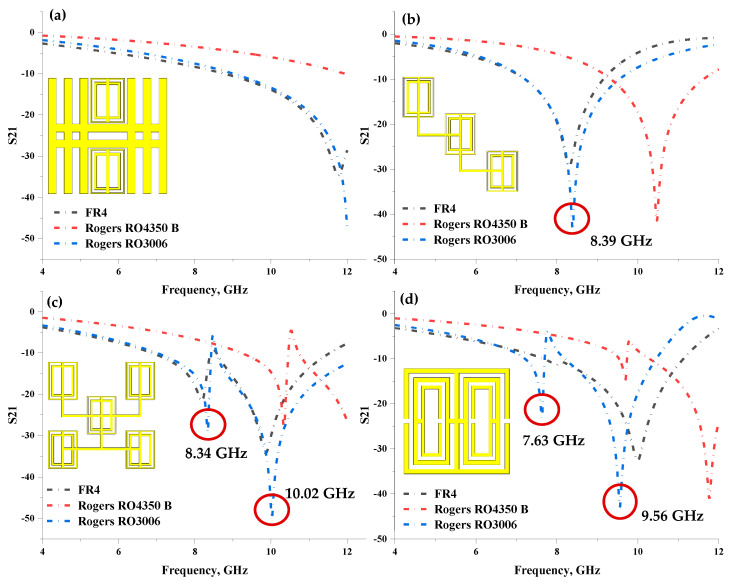
Transmission coefficient, S21 of: (**a**) Design 1, (**b**) Design 2, (**c**) Design 3, and (**d**) Design 4.

**Figure 2 materials-16-04566-f002:**
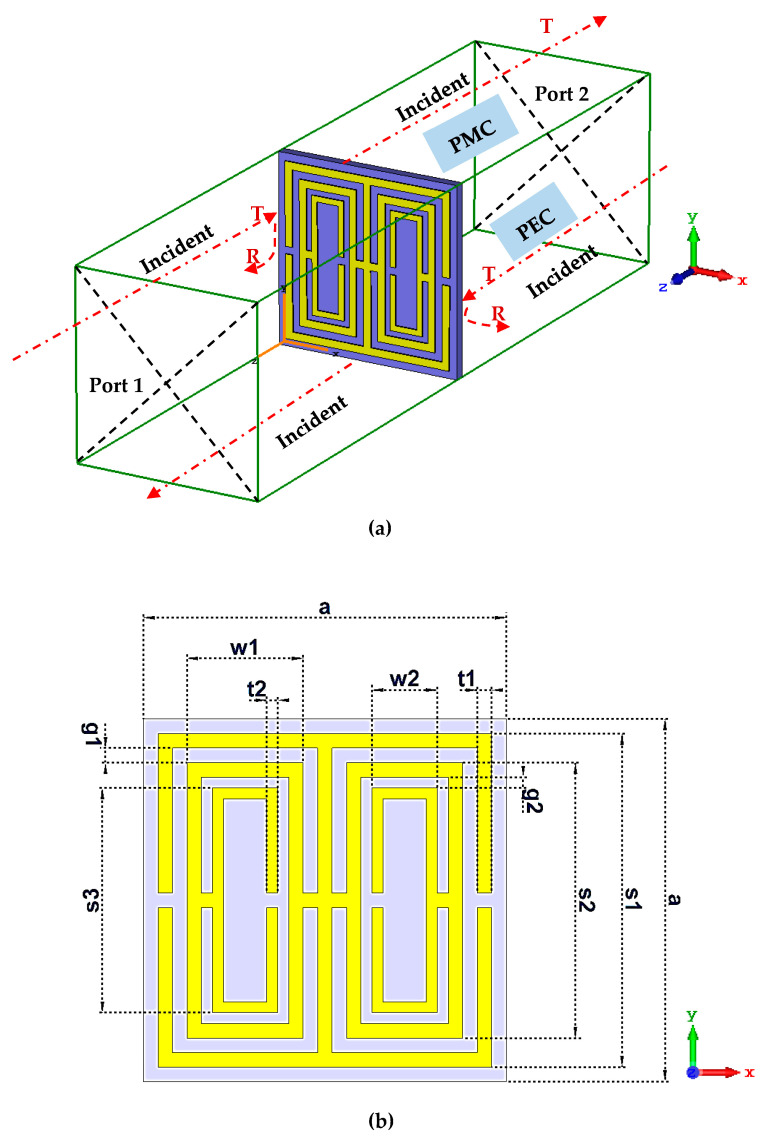
(**a**) Set up of boundary condition of the proposed symmetric metamaterial design, and (**b**) proposed unit cell metamaterial design.

**Figure 3 materials-16-04566-f003:**
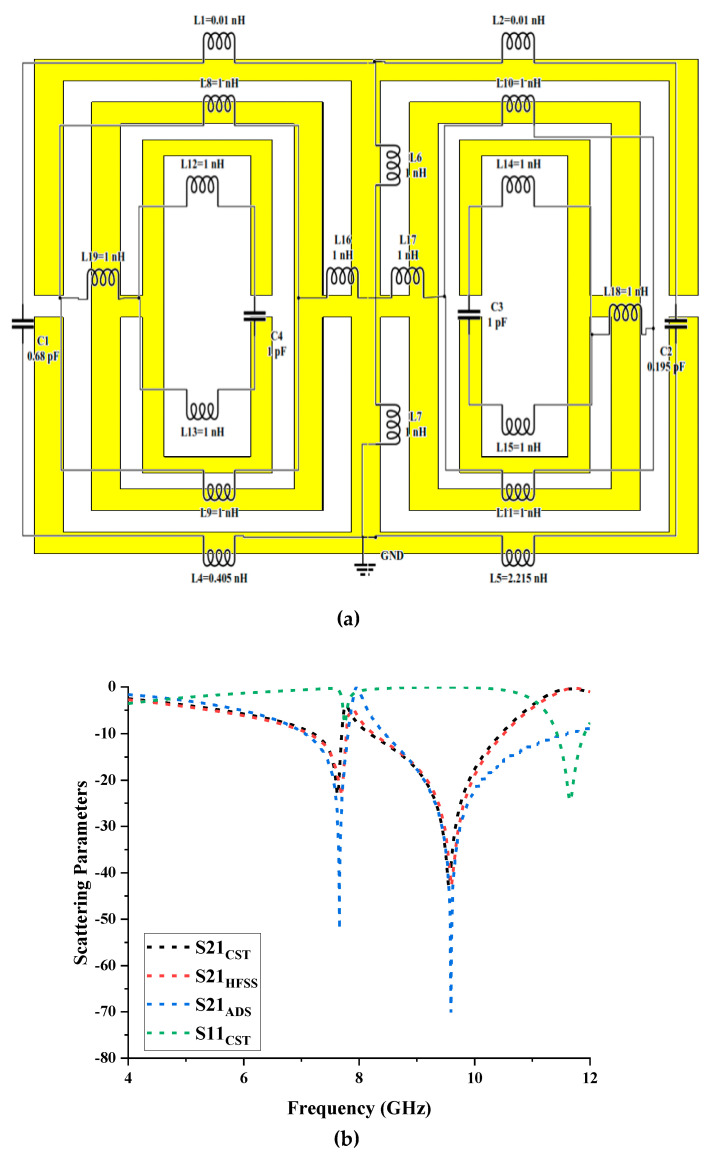
(**a**) Equivalent circuit, and (**b**) reflection and transmission coefficients of proposed metamaterial in CST, HFSS, and ADS software.

**Figure 4 materials-16-04566-f004:**
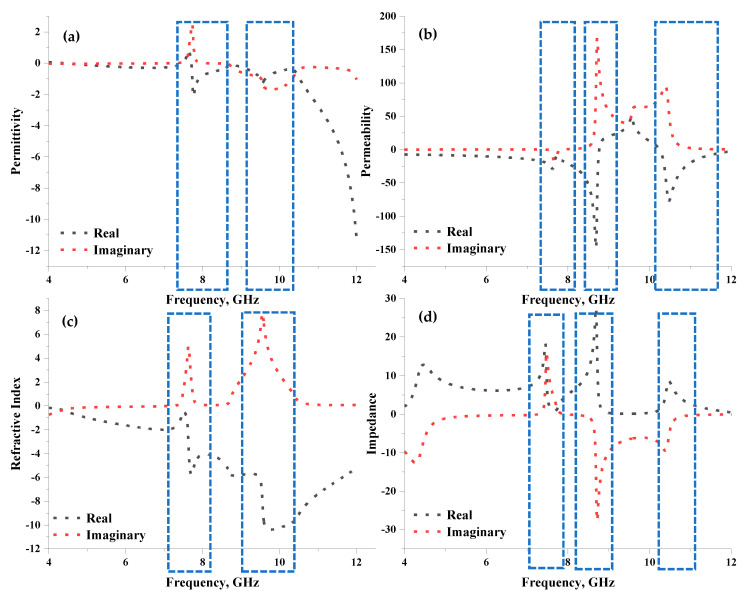
Effective medium parameters of the proposed metamaterial: (**a**) permittivity, (**b**) permeability, (**c**) refractive index, and (**d**) impedance.

**Figure 5 materials-16-04566-f005:**
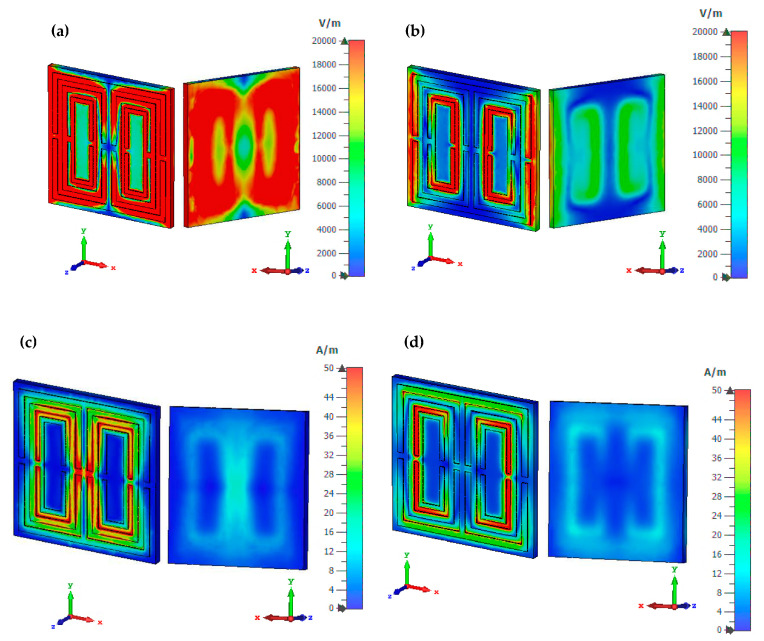
Field distribution of the proposed design: (**a**) electric at 7.63 GHz, (**b**) electric at 9.56 GHz, (**c**) magnetic at 7.63 GHz, and (**d**) magnetic at 9.56 GHz.

**Figure 6 materials-16-04566-f006:**
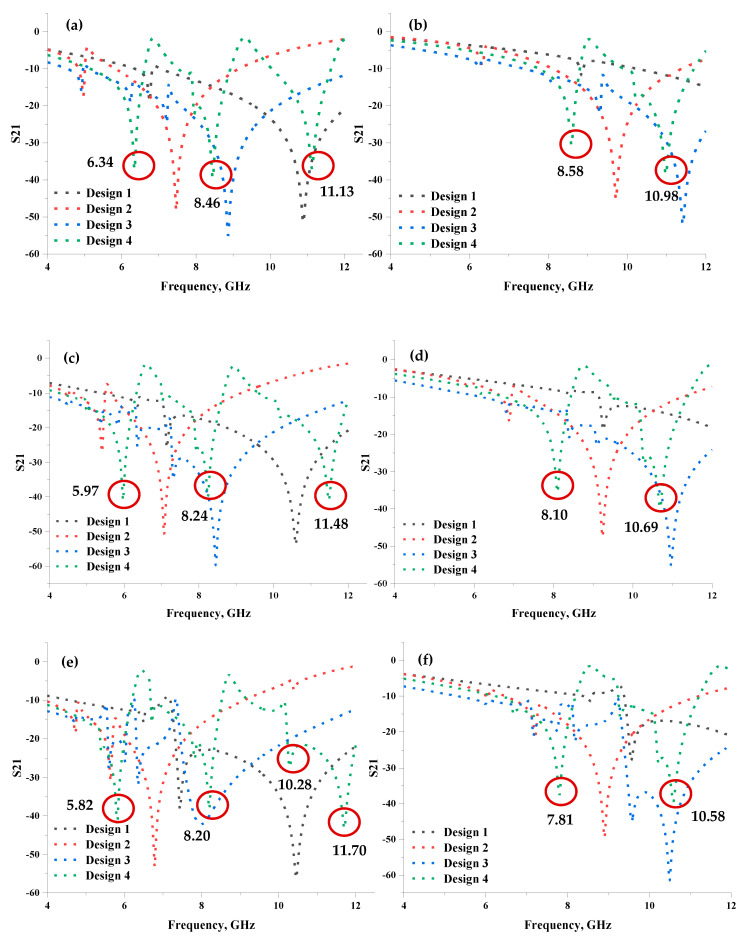
Transmission coefficient of the multi-layered metamaterial structure: (**a**) double-layer Rogers RO3006, (**b**) double-layer Rogers RO4350B, (**c**) triple-layer Rogers RO3006, (**d**) triple-layer Rogers RO4350B, (**e**) quadruple-layer Rogers RO3006, and (**f**) quadruple-layer Rogers RO4350B.

**Figure 7 materials-16-04566-f007:**
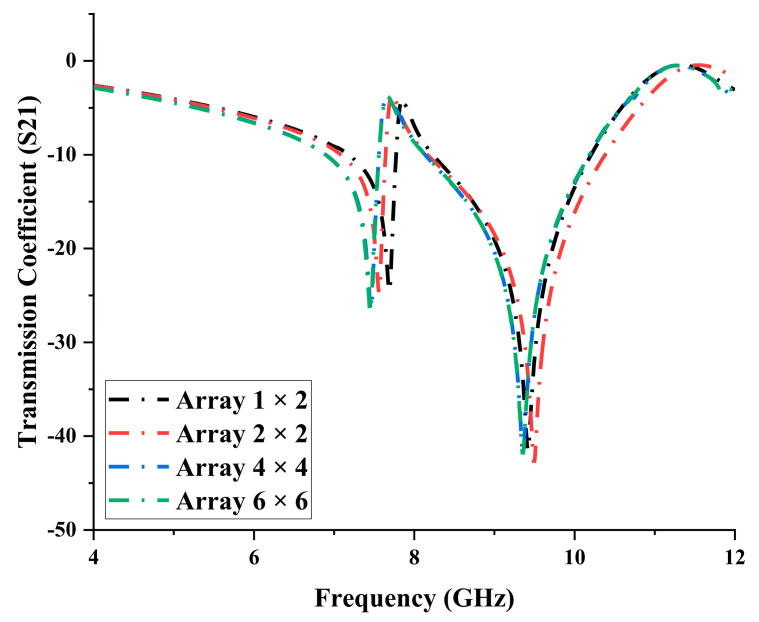
Transmission coefficients of array metamaterial structure, such as 1 × 2, 2 × 2, 4 × 4, and 6 × 6.

**Table 1 materials-16-04566-t001:** Resonant frequencies of four distinct metamaterial designs using three types of substrate materials.

Substrate Materials	Thickness (mm)	Number Resonant Frequencies	
FR4	1.6	Design 1	1 (11.77 GHz)
		Design 2	1 (8.30 GHz)
		Design 3	2 (8.18, 9.87 GHz)
		Design 4	1 (10 GHz)
Rogers RO4350B	0.168	Design 1	0
		Design 2	1 (10.47 GHz)
		Design 3	1 (10.34 GHz)
		Design 4	1 (11.77 GHz)
Rogers RO3006	0.25	Design 1	0
		Design 2	1 (8.39 GHz)
		Design 3	2 (8.34, 10.02 GHz)
		Design 4	2 (7.63, 9.56 GHz)

**Table 2 materials-16-04566-t002:** The dimension of the proposed metamaterial in mm.

Description (mm)			
Substrate length and width, a	5.0	Semi ring 2 width, t_2_	1.5
Square ring length, s_1_	4.6	Semi ring 1 width, w_1_	1.6
Semi ring 1 length, s_2_	3.8	Semi ring 2 width, w_2_	0.9
Semi ring 2 length, s_3_	3.1	Gap between ring, g_1_	0.2
Square ring width, t_1_	2.0	Gap between ring, g_2_	0.15

**Table 3 materials-16-04566-t003:** Comparison of previously published articles with the proposed design.

References	Dimension (mm)	Substrate Material	Operating Frequency (GHz)	Resonant Frequency/Application	Type
[25]	Unit cell: 10 × 10	FR-4	1–15	11.92 GHz (X-Band)	Negative index
[26]	Unit cell: 10 × 10	FR-4	2–14	3.53, 7.60, 12.94 GHz (S, C, Ku-bands)	Left-handed
[10]	Unit cell: 10 × 10	FR-4	2–14	7.32, 12.02 GHz (C- and X-bands)	Left-handed
[16]	Unit cell: 10 × 10	FR-4	8–14	7.69, 8.47, 13.14, 12.04 GHz (C-, X-, and Ku-bands)	Double-negative
[27]	Unit cell: 9.6 × 9.6	FR-4	1–15	1.54, 8.46, 12.33 GHz (L-, X-, Ku-bands)	Negative index
[28]	Array cell: 180 × 180	3 types of CI-PEEK composite	12–18	Absorber	Wide-angle
[29]	Array cell: 200 × 200	Multiple	0–20	Reflection coefficient: 5, 15 GHz (C- and X-bands)	Wide-angle
[30]	Array cell: 22.86 × 10.16	δ = 0.0013 ε = 2.56	7–14	Five response	Broadband
Proposed	Unit cell: 5 × 5 × 0.285 Multi-layer: 5 × 5 × 0.570	Rogers RO3006	4–12	Unit cell: 7.63, 9.56 GHz (C-, X-bands) Multi-layer: 6.34, 8.46, 11.13 GHz (C-, X-bands)	Left-handed

## Data Availability

All the data are available within the manuscript.

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
