# Peer review of "Compact Multi-Layered Symmetric Metamaterial Design Structure for Microwave Frequency Applications"

_materials, 2023, doi:10.3390/ma16134566_

Round 1

Reviewer 1 Report

1. This manuscript focuses on a multi-layered symmetric metamaterial design for C- and X-band applications, however, I think about that the manuscript lacks novelty in design, definite research objectives, and meaningful research conclusions. In addition, the readability of the English expression is also poor.
2. There are many similar research results reported with definite research goals and better results, however, the authors of this manuscript doesn't seem to have paid enough attention to them in this manuscript.
3. The author did not use any new materials in the structural design and did not make any breakthroughs in the very simple simulation results.
4. As a research article, at least two research methods should be used during the studies to mutually verify the results, however, there is only simple numerical simulations in this manuscript.
5. Detailed comparisons of physical properties should be provided to demonstrate their advantages, novelties and breakthrough.
6. The process of parameter optimizations should be well shown to demonstrate their design process.
7. The logical structure and the English expression of the article is also need to be improved.
8. Based on the above points of view, I cannot agree with the acceptance of this manuscript.

poor

Author Response

As attached.

Reviewer 2 Report

-The paper presents compact designs using multilayered structures. They have used different unit cells to accomplish the task. left-handed metamaterial was used in the process. The outcome was a compact structure with multiple resonances as was contemplated. the following are some points to be considered. 
There is no experimental verification of the structures. The paper is based on simulations, and while experiments significantly improve, the scientific outcome of the report in the C and X bands.

-There is no need for comparing several codes (Fig. 3) to justify the results. Rather, experimental data and only one code do the job.

-FR4 is lossy. A comment on the losses and efficiency is expected when FR4 and other substrates are employed in the designs.

-The authors have used similar notations for "permeability" and loss tangent. This makes confusion. Please change the loss tangent notation in lines 118,119,...

-The paper needs careful editing. Few Examples:

"Resonance frequency" and similar statements should change to "Resonant frequency" throughout the paper.

Line-33: "which not available"

Line 122: 0.168mm, (a space is needed)

Line 527: "On the hand"

-Line 530: "Between the range of 8 to 10 GHz.."

...

"Resonance frequency" and similar statements should change to "Resonant frequency" throughout the paper.

Line-33: "which not available"

Line 122: 0.168mm, (a space is needed)

Line 527: "On the hand"

-Line 530: "Between the range of 8 to 10 GHz.."

...

Author Response

As attached.

Reviewer 3 Report

Design of metamaterials with appropriate physical and electromagnetic properties has high relevance not just for the science but also for the practice. In this context, properties related to the attenuation, refractive index, absorption efficiency, or power transformation could be considered as key parameters. For practical purposes, the characteristics of EM-metamaterial interaction at microwave frequency ranges have high importance. Manuscript has a special focus on the investigation of symmetrical metamaterials for weather monitoring. Therefore, the topic of the manuscript can be considered as novel relevant and interesting for the readers. The manuscript has a proper structure. Introduction summarizes well the relevance of the study and the specific research motivations. The metamaterial design analysis is described clearly. Figures and tables are informative. Section 5 is dedicated for comparison with reference work. The mnauscript contains novel, significant and valuable results, that are discussed clearly and demonstrated well in tables and figures.

Comments, suggestions:

Please use equation for Eq.5, unify the style of equations and mathematical formulas.

Table 3 is rather a figure. Please improve the visibility of scales, text.

Section 3 and 4 are poorly referenced.

Author Response

As attached.

Round 2

Reviewer 1 Report

accept

Read carefully again and again.

Author Response

As attached.

Reviewer 2 Report

-Although the paper uses several codes, experimental data is crucial in these types of work. In this frequency range, it is not a difficult task. Experimental verification is recommended.

-The paper still suffers from typos and grammatical errors and needs careful checking.

"resonance frequency" should be changed to "resonant frequency" throughout the manuscript.

Author Response

As attached.
